# Treatment of Monochlorobenzene from Polymers Process through Electrochemical Oxidation

**DOI:** 10.3390/polym16030340

**Published:** 2024-01-26

**Authors:** Baiqi Wang, Yanmin Yue, Siyi Wang, Yu Fu, Chengri Yin, Mingji Jin, Yue Quan

**Affiliations:** 1Department of Agricultural Resources and Environment, Yanbian University, Yanji 133002, China; wang17743067721@outlook.com (B.W.); yym3225244730@hotmail.com (Y.Y.); 15640621277@163.com (S.W.); jinmingji@ybu.edu.cn (M.J.); 2Department of Chemistry, Yanbian University, Yanji 133002, China; fuyu@ybu.edu.cn (Y.F.); cryin@ybu.edu.cn (C.Y.); 3Department of Geography and Ocean Sciences, Yanbian University, Hunchun 133300, China

**Keywords:** polymers process, chlorobenzenes, waste gas, electrochemical oxidation

## Abstract

With the rapid development of the economy and the demands of people’s lives, the usage amount of polymer materials is significantly increasing globally. Chlorobenzenes (CB_S_) are widely used in the industrial, agriculture and chemical industries, particularly as important chemical raw materials during polymers processes. CB_S_ are difficult to remove due to their properties, such as being hydrophobic, volatile and persistent and biotoxic, and they have caused great harm to the ecological environment and human health. Electrochemical oxidation technology for the treatment of refractory pollutants has been widely used due to its high efficiency and easiness of operation. Thus, the electrochemical oxidation system was established for the efficient treatment of monochlorobenzene (MCB) waste gas. The effect of a single factor, such as anode materials, cathode materials, the electrolyte concentration, current density and electrode distance on the removal efficiency (RE) of MCB gas were first studied. The response-surface methodology (RSM) was used to investigate the relationships between different factors’ conditions (current density, electrolyte concentration, electrode distance), and a prediction model was established using the Design-Expert 10.0.1 software to optimize the reaction conditions. The results of the one-factor experiments showed that when treating 2.90 g/m^3^ MCB gas with a 0.40 L/min flow rate, Ti/Ti_4_O_7_ as an anode, stainless steel wire mesh as a cathode, 0.15 mol/L NaCl electrolyte, 10.0 mA/cm^2^ current density and 4.0 cm electrode distance, the average removal efficiency (RE), efficiency capacity (EC) and energy consumption (Esp) were 57.99%, 20.18 g/(m^3^·h) and 190.2 (kW·h)/kg, respectively. The results of the RSM showed that the effects of the process parameters on the RE of MBC were as follows: current density > electrode distance > electrolyte concentration; the interactions effects on the RE of MBC were in the order of electrolyte concentration and current density > current density and electrode distance > electrolyte concentration and electrode distance; the optimal experimental conditions were as follows: the concentration of electrolyte was 0.149 mol/L, current density was 18.11 mA, electrode distance was 3.804 cm. Under these conditions, the RE achieved 66.43%. The response-surface variance analysis showed that the regression model reached a significant level, and the validation results were in agreement with the predicted results, which proved the feasibility of the model. The model can be applied to treat the CB_S_ waste gas of polymer processes through electrochemical oxidation.

## 1. Introduction

Numerous kinds of polymer materials have emerged and been popularized in various industries, as well as in people’s common life over the past few decades [1,2]. The rapid development of the synthesis of organic polymer materials, including electric and electronic devices, fire-fighting foams, photo imaging, hydraulic fluids and metal plating, have played important roles but also caused serious environmental pollution [3,4]. Nonlinear polymers with star, branched, dendritic and cyclic structures were synthesized for polymer chemistry to achieve progress in the field [5]. These unique polymers are called advanced functional polymers with topology. The topologies often lead to changes in the physical and chemical properties of solutions, bulk or melts, which bring potential applications in many fields, such as energy harvesting, storage, wireless communication, biomedicine, oil/water or gas separation and intelligent bionics [6]. Thus, the environmental pollution caused by polymers has raised concerns as the usage of polymers continues to increase. Microplastics (MP) are defined as plastic products less than 5 mm in size and mainly come from personal care and cosmetic products (PCCPs). They are synthetic polymers with properties such as a higher durability, high strength, bio-inertness and a barrier [7]. The lethal impacts of MPs include malnutrition and starvation, retardation of reproduction and internal damage to tissues and organs of faunal groups when MPs are mistakenly eaten. MPs also cause the transfer of chemicals and toxins via MPs through ingestion [8]. As fluorinated polymers, fluoroplastics have high molecular weight polymers (>100,000 Da), mainly used in semiconductor manufacturing, glass production, fertilizer production and metal processing [9]. The World Health Organization (WHO) classifies fluoride, along with arsenic and nitrate, as one of the contaminants of drinking water that can cause lots of health problems [10,11]. The permissible threshold concentration of fluoride in drinking water is prescribed as 1.5 mg/L according to the Environmental Protection Agency and WHO guidelines [12]. Polyfluoroalkyl substances from synthetic organic fluoride are widely used and have strong C–F bonds. For these reasons, they are difficult to degrade and easily accumulate in tissues and organs with a high protein content, such as the blood, liver and kidneys [13].

Chlorobenzenes (CB_S_) are an important organic chemical material and have been widely used in polymer materials to produce polymer plastics and polymer resin. At the same time, CB_S_ have been used as a solvent in polymer rubber chemicals and polymeric coatings using the single thick layer technique [14]. CB_S_ possess stable chemical properties, have a high volatility and are water-insoluble, which results in them being difficult to degrade. CB_S_ are easily released into the environment and cause serious air pollution, water pollution and soil pollution, particularly through the formation of dioxins, photochemical smog, ozone layer deterioration and haze [15]. In addition, CB_S_ are toxic to humans and accumulate in the human body, which damages the liver, kidney, immune system and nervous system and can lead to cancer, deformity and mutation [16]. Thus, CB_S_ were listed as the priority control as typical persistent VOCs in many countries and regions, and their treatment has emerged as a critical issue worldwide.

Conventional technologies such as absorption, adsorption, thermal combustion, catalytic oxidation, catalytic steam reforming, photocatalytic decomposition, catalytic hydrochlorination and biofiltration have been used in the treatment of CB_S_ [17]. However, these technologies have the disadvantages of poor RE, secondary pollutants, high energy consumption (Esp) and high costs. In recent years, advanced oxidation processes based on electrochemical advanced oxidation processes (EAOPs) are considered as an ideal alternative for the removal of refractory pollutants. EAOPs have attracted increasing attention for their advantages, such as their small device size, mild reaction conditions, fewer additional chemicals, easy operation, no secondary pollutions and effective treatment effect and can be considered as an efficient method for treating highly polluted wastewater [18]. In the treatment process, a direct current is applied to an electrolyte solution to oxidize and reduce the target object at the anode and cathode poles, respectively. At the same, hydroxyl free radicals (·OH) and other strong oxidants are generated in the reaction process, which have a high redox potential [19], so that EAOPs convert organic compounds or POPs into harmless substances, such as simple inorganic molecules like carboxylic acids, CO_2_ and H_2_O [20]. A concentration of 30 mg/L bisphenol A can be completely removed in 5.0 h with a SnO_2_/MWCNT anode by applying a 20 mA/cm^2^ current density, 0.1 M sodium sulfate electrolyte, at pH 4.0 [21]. Using a Ti/SnO_2_-Sb/La-PbO_2_ porous Ti plate as the anode, 1 mg/L chloroisopropyl phosphate was oxidized; the reaction rate constant was 0.0502 min^−1^ and the degradation efficiency was 98.4% when the current density was 10 mA/cm^2^ [22]. The carbon-coated Ti_4_O_7_ electrode was applied for degrading phenol-containing wastewater (100 mg·L^−1^) under the condition of 0.1 M NaCl electrolyte, 20 mA cm^−2^ current density and 60.0 min, a complete degradation of phenol (100%), high TOC removal ratio (68.9%) and low Esp (113.8 kWh·kg^−1^ TOC) were achieved, respectively [23].

Electrochemical oxidation technology is widely used in the wastewater treatment field; however, little information is reported about organic waste gas, especially for polymers processes. Therefore, in this study, the electrochemical system of waste gas was constructed, where monochlorobenzene (MCB) waste gas was the target. The influences of important parameters such as positive materials, cathode materials, electrolyte concentration, current density and plate spacing on RE and Esp were studied. The response-surface method was used to optimize the process parameters and explore the best process conditions. Our work is devoted to bringing new insights into refractory waste gas treatment from polymers processes by electrochemical advanced oxidation technology.

## 2. Materials and Methods

### 2.1. Construction of Electrochemical System

The experimental system, as shown in Figure 1, consisted of four parts: an air pumping system, a VOC generator (FD-PG, Suzhou Furan De Experimental Equipment Co., Ltd., Suzhou, China), an electrolytic cell and a DC power supply. The electrolytic cell is made of plexiglass (height: 300.0 mm, length: 120.0 mm, width: 120.0 mm). The inlet waste gas and microporous aerator device were provided at the bottom of the electrochemical reactor. The middle part of the reactor is dedicated to the electrochemical oxidation; the anode and cathode (effective size was 100 × 80 cm) were placed in parallel on a sieve plate with an adjustable distance, which is connected with the DC power supply (KPS-3005D, Shenzhen Zhaoxin Electronic Instrument Equipment Co., Ltd., Shenzhen, China) by an electrode holder with a conductor. MCB waste gas is produced by an injection pump and VOC generator. The air pump produces air, which dilutes the MCB waste gas. The gas flow rate is regulated by a mass flow controller (GSC-B9SB-BB26, Vögtlin Instruments AG, Aesch, Switzerland). The waste gas flows into the microporous aerator at the bottom, and it is then treated by electrochemical oxidation; untreated waste gas was discharged at the top outlet.

### 2.2. Experimental Method

#### 2.2.1. Single-Factor Experiment 

The effects of the anode material, cathode material, electrolyte concentration (0.05, 0.10, 0.15, 0.20, 0.30 mol/L), current density (2.0, 4.0, 10.0, 20.0, 30.0 mA/cm^2^) and electrode distance (3.0, 4.0, 5.0 cm) on MCB waste gas removal were investigated. Each degradation experiment was repeated at least three times to ensure the reproducibility of the data. Experimental procedure: Firstly, the VOC generator was turned on; then, the DC power supply was turned on to adjust the voltage and keep the constant current mode until the inlet concentration was stable. Secondly, the gas concentration was measured by gas chromatography, and the electrolytic voltage was recorded. Lastly, the RE, EC, Esp and ACD were calculated:
RE = (C_in_ − C_out_)/C_in_ × 100%
EC = (C_in_ − C_out_)Q/V
Esp = UI/(C_in_ − C_out_)Q
CD = I/A
where RE represents the removal efficiency, %; C_in_ and C_out_ represent the inlet and outlet concentrations, mg/m^3^; EC is the efficiency capacity, g/(m^3^·h), Q is the gas flow, m^3^/h; V is the volume of the electrolyte, m^3^; Esp is the energy consumption, kW·h/kg; U is the reaction voltage, V; I is the current, A; CD is the average current density, mA/cm^2^; and A is the effective area of the anode, cm^2^.

#### 2.2.2. Response-Surface Optimization Experiment

Based on the results of single-factor experiments, the response-surface optimization experiments were based on the RE of MCB waste gas and economic factors, and a Box–Behnken design experiment was carried out using Design-Expert 10.0.1 software with three factors (electrolyte concentration *X*_1_, current density *X*_2_, electrode distance *X*_3_) and three levels. The quadratic polynomial regression equation is obtained. The levels and codes of experimental factors of the response surface are shown in Table 1.

### 2.3. Electrode Material

The anode materials used in the experiment were Ti/Ti_4_O_7_ (Zhengyinghao Metal Material Trading Company, Hefei, China), BDD (Zhengyinghao Metal Material Trading Company, Hefei, China), Ti/RuO_2_-IrO_2_ (Baoji Rui Cheng Titanium Metal Co., Ltd., Baoji, China) and Ti/PbO_2_ (Xian Taijin New Energy and Sci-Tech Co., Ltd., Xi’an, China). The cathode materials were titanium plate (Baoji Rui Cheng Titanium Metal Co., Ltd., Baoji, China), graphite plate (Qinghe County Dingyuan Metal Products Co., Ltd., Xingtai, China), stainless steel mesh (Anping County Tianhong Wire Mesh Factory Carbon Felt, Hengshui, China) and carbon felt (Anhui new poly carbon fiber Co., Ltd., Chuzhou, China).

### 2.4. Analysis Method

The concentration of the MCB waste gas was analyzed by a gas chromatograph (7890B, Agilent, Santa Clara, CA, USA) with a FID detector and HP-5 column (19091J-413, Agilent). The heating procedure is as follows: the initial temperature of the column box is 40.0 °C for 2.0 min, and increased to 200.0 °C with a rate of 30.0 °C/min. Nitrogen is the carrier gas with a flow rate of 10.0 mL/min, hydrogen (purity ≥99.999%) is the fuel gas with a flow rate of 30.0 mL/min, the temperature of the inlet was 150.0 °C, injections were made in the split mode with a split of 20 and the volume of injector gas was 1000.0 μL. Under these conditions, the retention time of CB was 2.21 min.

### 2.5. Electrochemical Test

The electrochemical activity and stability of the Ti/Ti_4_O_7_ electrode were investigated by electrochemical tests. The three electrode systems were carried out in the electrochemical workstation (CHI 660E, Shanghai, China). The Ti/Ti_4_O_7_, platinum and saturated calomel electrode (SCE) were chosen as the working, counter and reference electrodes, respectively. A linear voltammetry curve (LSV) was performed in 40 mmol·L^−1^ Na_2_SO_4_, 40 mmol·L^−1^ Na_2_SO_4_ and 10 mg^−1^ MCB solution, respectively. The scanning speed was set to 50 mV·s^−1^, and the curve from 0 to 3 V_SCE_ was recorded. According to the electric potential corresponding to the inflection point of the polarization curve, the oxygen evolution potential (OEP) could be obtained. Cyclic voltampere (CV) curves were tested from −2 to 3 V_SCE_.

## 3. Results

### 3.1. Effect of Anode Materials

The selection of anode materials is the key point of EAOPs. Anode materials as a working electrode are the core of electrochemical oxidation processes in the current efficiency of anodic oxidation and RE [24]. For the evaluation of anode material, the effects of the anode material on the RE, EC and Esp of MCB waste gas by electrochemical oxidation are shown in Figure 2. In this study, BDD, Ti/PbO_2_, Ti/Ti_4_O_7_ and Ti/RuO_2_-IrO_2_ were selected as anode materials, and the titanium plate as the cathode, under the processing conditions of 0.10 mol/L NaCl solution, 10.0 mA/cm^2^ current density and 4.0 cm electrode distance. As shown in Figure 2, the RE, EC and Esp exhibited little trend for treating MCB under four different anode materials. The RE of Ti/BDD anode was the highest, followed by Ti/Ti_4_O_7_, Ti/RuO_2_-IrO_2_ and Ti/PbO_2_, while the EC and Esp of the Ti/Ti_4_O_7_ anode were significantly better than for Ti/RuO_2_-IrO_2_ and Ti/PbO_2_. The BDD electrode has a high oxygen evolution potential and high catalytic activity. In the reactions, a large number of hydroxyl radicals, peroxides, ozone and other strong oxidizing substances were generated, which can effectively degrade organic pollutants [25]. However, BDD electrodes have the disadvantage of high cost, which limits the application range in treating pollutants and needs to be further improved. Traditional electrode materials exhibit several limitations, such as a short service life, poor removal effect and high cost, including graphite electrodes, metal electrodes and dimensionally stable anodes [26]. As a new electrode material, Ti/Ti_4_O_7_ has been widely regarded as a promising electrode material for electrochemical applications [27]. Ti/Ti_4_O_7_ has advantages such as a 15.0 s/cm conductivity, a potential for oxygen evolution up to 3.0 V, electrical conductivity up to 15.0 s/cm and a wide potential window [28]. Meanwhile, Ti/Ti_4_O_7_ resists strong acid and alkali corrosion, and has stable chemical properties and a lower cost [27]. In conclusion, considering cost, RE and Esp, Ti/Ti_4_O_7_ was selected as the next anode material to save energy and green development. The RE, EC and Esp reached 50.21%~51.87%, 27.26 g/(m^3^·h) and 305.27 (kW·h)/kg in 10.0~50.0 min.

### 3.2. Effect of Cathode Materials

Cathodes are an important component during electrochemical oxidation reaction for hydrogen evolution reaction occurs. Few secondary reaction occurred in higher hydrogen evolution potential, and for the reduction reaction of cathodes have low contribution to treat pollutants. For these reasons, most studies have paid less attention to optimizing the cathode. But, the cathode reduces the oxygen produced by the anode to H_2_O_2_, which produces ·OH and improves the treatment effect. Thus, in order to systematically study cathode materials’ effect on the RE, EC and Esp of MCB gas, titanium plate, graphite plate, stainless steel mesh and carbon felt were used. The other parameters of electrochemical oxidation were Ti/Ti_4_O_7_ as the anode, 0.10 mol/L NaCl, 10.0 mA/cm^2^ current density, 4.0 cm electrode distance and electrolysis for 10.0–50.0 min. The experimental results are shown in Figure 3.

As can be seen from Figure 3, the RE, EC and Esp of the four cathode materials presented great differences during 10.0–50.0 min. When the cathode material is a stainless steel mesh, the RE and EC of the MCB gas were the best and reached 55.85% and 15.81 g/(m^3^·h), respectively. Esp showed the lowest value for 279.62 (kW·h)/kg. The stainless steel mesh was selected as the cathode material of the next experiment for its good treatment efficiency and low Esp.

### 3.3. Effect of Electrolyte Concentration

The main functions of electrolytes are to improve the conductivity of solutions, promote the transfer of electrons and offer different inorganic oxidants, resulting in pollutant removal [29]. In this study, NaCl was selected as an electrolyte to generate Cl_2,_ which can be converted to ClO^−^ with strong oxidation, so the RE of MCB waste gas could be accelerated. The specific process was as follows [30]:
2Cl^−^ → Cl_2_(aq) + 2e^−^Cl_2_(aq) + H_2_O → HClO + Cl^−^ + H^+^ (the chloride ion cycle)HClO ↔ ClO^−^ + H^+^ (in equilibrium)O_2_ + 2H^+^ + 2e^−^ → H_2_O_2_


The influence of NaCl electrolyte concentration (0.05, 0.10, 0.15, 0.20 and 0.30 mol/L) on the RE, EC and Esp of MCB gas were investigated, when Ti/Ti_4_O_7_ was used as the anode, stainless steel mesh as the cathode, 10.0 mA/cm^2^ current density and 4.0 cm electrode distance for 10.0–50.0 min of electrolysis. The results are shown in Figure 4; lower or higher electrolyte concentrations also have a great influence on the removal effect. When the electrolyte concentration was 0.15 mol/L, the best RE was achieved and the average RE and EC were 58.55% and 20.39 g/(m^3^·h), respectively. Meanwhile, the lowest Esp was 188.47 (kW·h)/kg. When 0.30 mol/L NaCl was used, the treatment effect was the worst. The reason is that when the electrolyte concentration is high, the interaction force between ions increases, causing a decrease in the ion movement rate and reaction rate [31]. At the same time, a large amount of Cl^−^ ions are absorbed on the electrode surface, which hinders the production of ·OH [31]. In order to save energy, a 0.15 mol/L NaCl electrolyte concentration was selected for the next experiment.

### 3.4. Effect of Current Density

Current density is an important operating parameter during electrochemical oxidation for the evaluation of the Esp and RE of MCB. Figure 5 showed the influence of the current density (2.0, 4.0, 10.0, 20.0, 30.0 mA/cm^2^) on the treatment efficiency. In this study, the anode material, cathode material, electrode distance and electrolyte concentration were Ti/Ti_4_O_7_, stainless steel net, 4.0 cm and 0.15 mol/L NaCl, respectively. It can be seen that the RE, EC and Esp increase with increasing current density. When the current density was 2.0, 4.0, 10.0, 20.0 and 30.0 mA/cm^2^, the corresponding average RE was 41.36, 48.53, 58.54, 64.69 and 66.65%, respectively. The reason is that the anode produces more ·OH with an increasing current density, which increases the electron transfer speed in the electrolyte and promotes chemical reactions in the solution, leading to RE increasing. But a higher current density causes a voltage increase and Esp increase. Especially, the average Esp of the 20.0 mA/cm^2^ and 30.0 mA/cm^2^ current density increases to 615.08 (kW·h)/kg and 1147.73 (kW·h)/kg, respectively, which is much higher than 188.55 (kW·h)/kg when the current density is 10.0 mA/cm^2^.

### 3.5. Effect of Electrode Distance

The effects of different electrode distances of 3.0, 4.0, 5.0 and 6.0 cm on the RE of MCB are shown in Figure 6, when Ti/Ti_4_O_7_ was used as the anode, stainless steel net as the cathode, 0.15 mol /L NaCl, 10.0 mA/cm^2^ and electrolysis for 50.0 min. It can be seen from Figure 6 that with the increase in electrode distance, the RE and EC exhibited an upward trend at first and then a downward trend. When the electrode distance was 4.0 cm, RE and EC were the best and Esp was the minimum. The reasons are as follows: with the increase of electrode distance, the mass transfer coefficient decreases, the tank voltage increases with the resistance increases, side reactions such as hydrogen evolution increase, which leads to a reduction in efficiency and an increase in Esp [32]; a shorter electrode distance causes the passivation of the anode electrode plate, Esp increases, the concentration polarization of the electrolyte is serious and the RE is reduced [33]; the electrode distance is small and the strong oxidizing products such as solvated electrons and ·HO produced by the anode are directly reduced by the cathode and cannot oxidize organic pollutants in time. In addition, small electrode distance also easily caused a short-circuit tendency.

When the electrode distance is 3.0, 4.0 and 5.0 cm, the average Esp is about 172.79, 171.03 and 381.02 (Kw·h)/kg. It can be seen that when the electrode distance is 4.0 cm, RE, EC and Esp are the best, so 4.0 cm was chosen as the best electrode distance.

### 3.6. Optimize Process Parameters with Response-Surface Methodology

#### 3.6.1. Response-Surface Experiment Results and Variance Analysis

The Box–Behnken design (BBD) was used for RSM in the experimental design and applied with three factors at three levels using Design-Expert 10.0.1 with the bounds of the factors (independent variables). Each independent variable was coded at three levels between −*α* and +*α* at the ranges determined by the preliminary experiments, where the independent variables were the electrolyte concentration (*X*_1_), 0.10–0.20 mol/L; current density (*X*_2_), 10–30 mA; and electrode distance (*X*_3_), 3.0–5.0 cm. The levels of independent variables and the experimental plan based on BBD are presented in Table 1. The experimental results of RSM are shown in Table 2; the RE of the process was evaluated by analyzing the responses. Experimental data were fitted to the quadratic polynomial regression equation from the following equations: *Y* (RE) = 65.794 + 0.27375*X*_1_ − 3.365*X*_2_ − 3.19125*X*_3_ + 2.0625*X*_1_*X*_2_ − 0.38*X*_1_*X*_3_ − 1.6575*X*_2_*X*_3_ − 1.9295 *X*_1_^2^ − 8.077 *X*_2_^2^ − 7.3195 *X*_3_^2^.

The ANOVA results of this quadratic model are presented in Table 3, where a larger *F*-value and smaller *p*-value can be used to test the significance of the correlation coefficient. Either F > F_0.01_ or *p* < 0.01 indicates that the factor has a significant influence or the adaptation of the modulus is significant. It can be seen from Table 3 that the *F*-value is 57.86 and the *p*-value is *p* < 0.0001 which mean the adaptations of the quadratic models are significant [34]. The lack of fit (*p* = 0.5454 > 0.05) is not significant which means the nonlinear relationship between the factors and the response value is significant for the regression equation. The correlation is good for R^2^_Adj_ = 0.9697; when R^2^_Adj_ − R^2^_Pred_ = 0.9697 − 0.9062 = 0.0635 < 0.2, CV = 2.00% < 10%, which indicates the module’s high reliability and the precision of the regression equation [35]. Therefore, the quadratic polynomial regression equation can be used to analyze and predict the optimal process of RE.

#### 3.6.2. Interaction Effect of Electrolyte Concentration and Current Density

The data in Table 2 were studied by dimension reduction to analyze the interaction effects of the electrolyte concentration, current density and electrode distance on the RE. The response surface and contour diagram are shown in Figure 7, Figure 8 and Figure 9.

As can be seen from Figure 7, the interaction between electrolyte concentration and current density has a parabolic distribution on the RE, and the longitudinal span is large and the contour line presents a significant oval, indicating that the interaction has a significant impact on the RE. When the electrolyte concentration is constant, the RE increases with the increase in current density, while the RE showed a downward trend when the current density was bigger than 20.0 mA/cm^2^. As a result of the larger current density, the greater number of conductive particles and ·OH increased the degradation of pollutants; the side reactions such as hydrogen evolution and oxygen evolution intensity happened as the current density reached a certain level, resulting in the RE of the pollutants decreasing [36]. When the current density is constant, the RE increases with the increase in the electrolyte concentration, but it decreases when the concentration is greater than 0.18 mol/L. The reason is the increase in the number of conductive particles with the increase in electrolyte concentration; however, a bigger electrolyte concentration increases the force of ions, thus the RE of MCB decreased [37].

Figure 8 shows the influence of electrolyte concentration and electrode distance on RE at a constant current density. The results showed that the curved surface of the electrode distance fluctuated greatly, and the influence of the electrode distance on the RE was more significant than that of the electrolyte concentration. With the increase in electrode distance, the RE of MCB increased at first and then decreased. This is because the smaller electrode distance leads to a smaller transfer resistance of the metal [38] and the larger electrode distance increases the equivalent resistance [39]; thus, a high current density improved the RE of MCB.

Figure 9 shows the influence of current density and electrode distance on the RE at a constant electrolyte concentration. It can be seen from Figure 9 that the contour diagram has a wide longitudinal span and shows an obvious oval shape. The greatest RE was achieved at a 15.0–20.0 mA/cm^2^ current density and 3.5–4.0 cm electrode distance.

#### 3.6.3. Verification of the Model

The experiment was repeated three times under an electrolyte concentration (*X*_1_) of 0.15 mol/L, current density (*X*_2_) of 18.11 mA/cm^2^ and electrode distance (*X*_3_) of 4.0 cm. The RE_S_ were 67.12%, 66.53% and 67.46%, respectively. The average RE was 67.04% under the same condition and the RE predicted by the model was 66.43%. The *t*-test value between the measured RE and the predicted RE was 1.126, which did not reach a significant level, indicating that the difference between the measured RE and the predicted RE was not significant. The predicted value was in good agreement with the measured value. The method can be used to optimize electrochemical oxidation. The results show that the response-surface method is highly accurate and reliable in predicting the optimal conditions for treating MCB. The method can be used to optimize the electrochemical oxidation of MCB, which can effectively avoid the blindness of the experiment.

### 3.7. Characterization of the Electrochemical of Electrode

The performance of the oxygen electrode was evaluated according to the LSV (Figure 10a). The OEP of the Ti/Ti_4_O_7_ electrode was 2.35 V vs. SCE, higher than Ti/RuO_2_-IrO_2_ (1.67–1.80 V) [40], Ti/PbO_2_ (1.82–1.97 V) [41] and the platinum electrode (1.3–1.6 V) [42]. The degradation reaction of organic pollutants on the anode was competitive with the oxygen evolution reaction. The oxygen evolution reaction was more difficult under a higher OEP, and a higher current efficiency of electrochemical oxidation treating organic matter was obtained. Therefore, a high OEP was conducive to promoting the occurrence of the degradation reaction, indicating that the Ti/Ti_4_O_7_ electrode had a better electrocatalytic performance than the other electrodes. When the electrode had a high OEP, reactive oxygen species were more likely to be generated. Meanwhile, the ·OH and oxygen atoms in the metal oxide lattice adsorbed on the electrode surface primarily reacted with pollutants, which reduced the occurrence of the oxygen evolution reaction and energy consumption in the whole reaction process. The first LSV test was performed, and a significant peak current was observed for 0.25 V. After the second and third LSV test, the peak current did not attenuate, which means that electrochemical oxidation occurred on the Ti/Ti_4_O_7_ anode surface, and the adsorption did not occur in the process.

Figure 10b displays the CV curves of the Ti/Ti_4_O_7_ electrode. The Ti/Ti_4_O_7_ electrode showed an obvious oxidation peak and reduction peak at 0.5 V (vs. SCE) and 0.25 V (vs. SCE). This is consistent with the LSV test results, which means that the electrochemical oxidation of MCB occurred on the Ti/Ti_4_O_7_ anode surface and the Ti/Ti_4_O_7_ electrode itself reacted in reduction and oxidation reactions. In conclusion, the Ti/Ti_4_O_7_ electrode had a higher electrocatalytic activity, a wide voltage window and a low adsorption performance, which were conducive to improving the degradation efficiency of organic pollutants.

## 4. Conclusions

In this study, an electrochemical oxidation system was constructed for treating MCB from polymers processes with a 0.40 L/min flow rate and 2.90 g/m^3^ concentration. When Ti/Ti_4_O_7_ was used as the anode and stainless steel wire mesh as the cathode, with 0.15 mol/L NaCl electrolyte, 10.0 mA/cm^2^ current density and 4.0 cm electrode distance, the average RE, EC and Esp values were 57.99%, 20.18 g/(m^3^·h) and 190.2 (kW·h)/kg, respectively. The experimental results showed that a high RE of MCB can be obtained at appropriate reaction conditions by electrochemical oxidation. The use of an experimental design (the RSM based on a Box–Behnken design) permitted the rapid screening of a large experimental domain for the optimization of RE ability of MBC. The fit of the model was checked by the determination coefficient (R^2^). In this case, the value of the determination coefficient (R^2^ = 0.9867) was indicated. The value of the adjusted determination coefficient (adjusted R^2^ = 0.9697) was also high, showing the high significance of the model. The optimized conditions for the highest RE (66.43%) of MCB are an electrolyte concentration of 0.15 mol/L, an electrode distance of 4.0 cm and a current density of 18.11. These results showed that electrochemical oxidation has an enormous potential to treat recalcitrant organic waste gas and to resolve the emission of industrial waste gas from polymers processes.

## Figures and Tables

**Figure 1 polymers-16-00340-f001:**
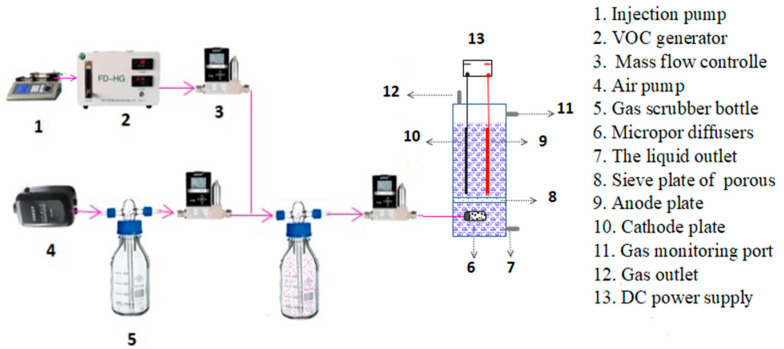
Experimental setup of electrochemical oxidation.

**Figure 2 polymers-16-00340-f002:**
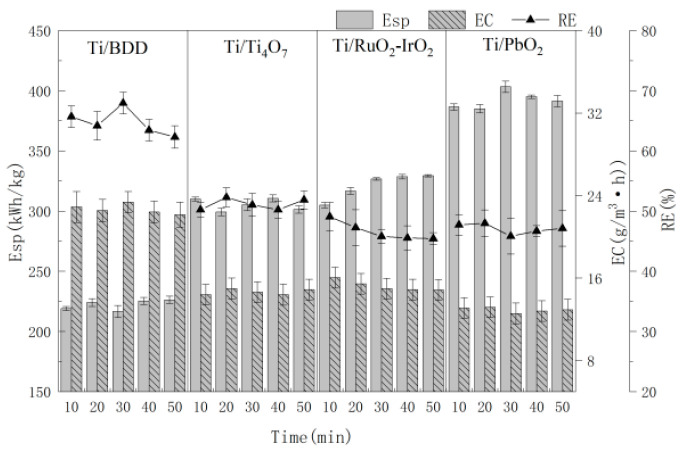
The effect of anode material on RE with time.

**Figure 3 polymers-16-00340-f003:**
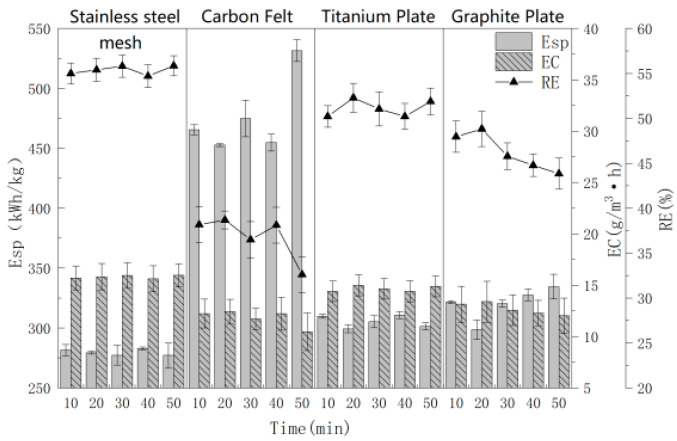
The effect of cathode materials on RE with time.

**Figure 4 polymers-16-00340-f004:**
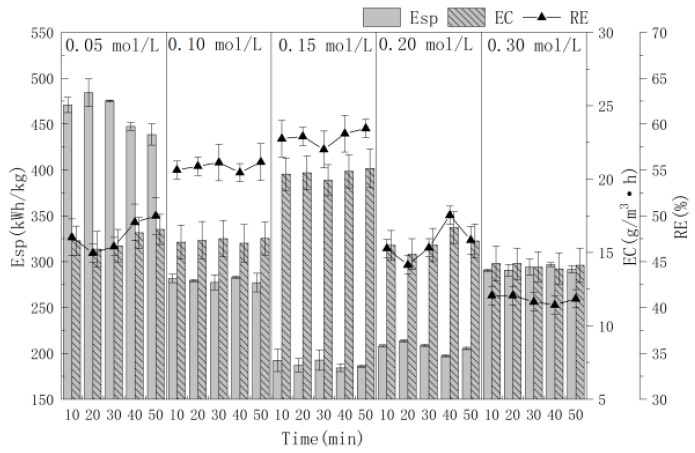
The effect of electrolyte concentration on RE with time.

**Figure 5 polymers-16-00340-f005:**
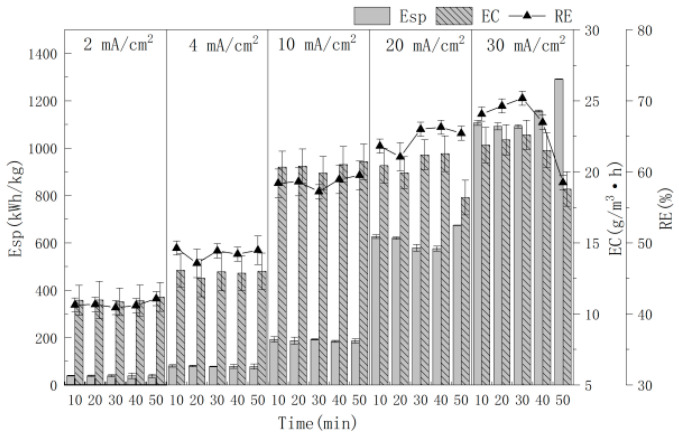
The effect of current density on RE with time.

**Figure 6 polymers-16-00340-f006:**
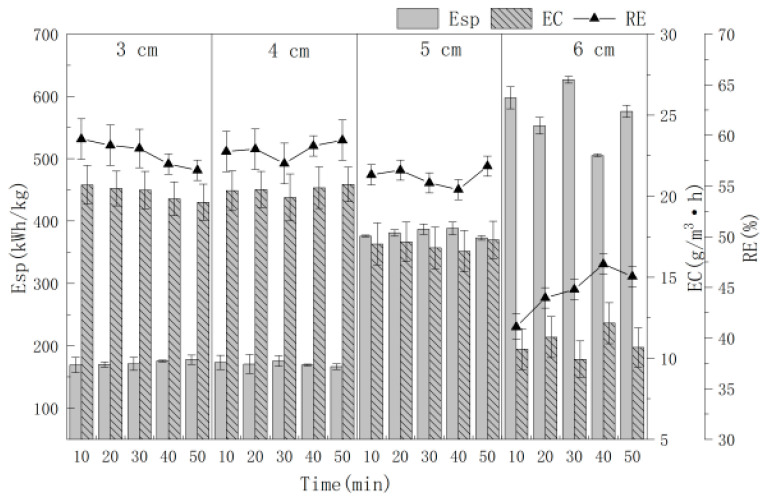
The effect of electrode distance on RE with time.

**Figure 7 polymers-16-00340-f007:**
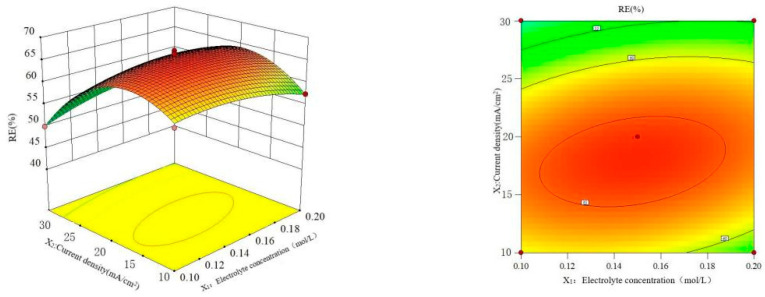
Interaction effect of electrolyte concentration and current on RE.

**Figure 8 polymers-16-00340-f008:**
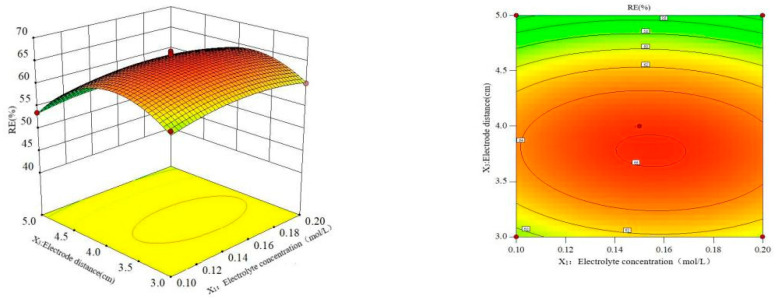
Effect of the interaction of electrolyte concentration and plate spacing on RE.

**Figure 9 polymers-16-00340-f009:**
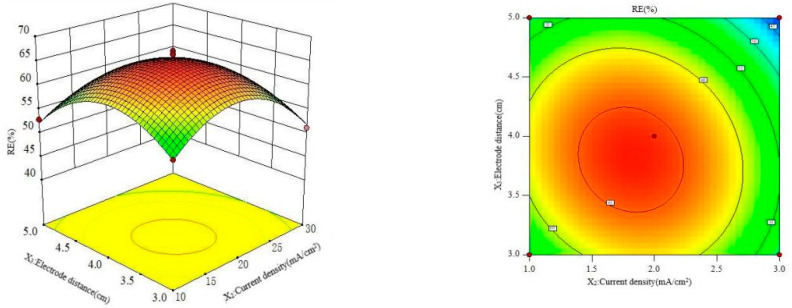
Interaction effect of current and plate spacing on RE.

**Figure 10 polymers-16-00340-f010:**
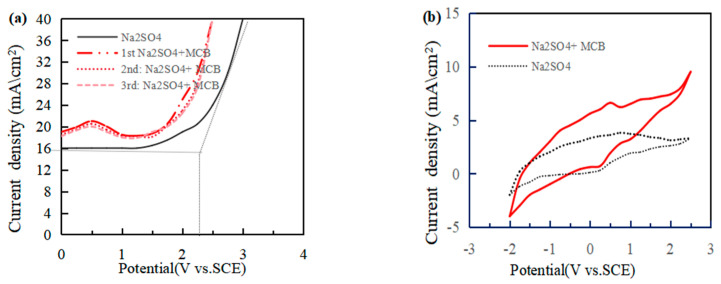
LSV (**a**) and CV (**b**) curves in Na_2_ SO_4_ solution and MCB solution by the Ti /Ti_4_O_7_ anode.

**Table 1 polymers-16-00340-t001:** Test factor level and coding of the response surface.

Code	Electrolyte Concentration *X*_1_ (mol/L)	Current Density *X*_2_ (mA/cm^2^)	Electrode Distance *X*_3_ (cm)
−1	0.10	10	3
0	0.15	20	4
1	0.20	30	5

**Table 2 polymers-16-00340-t002:** Experimental design and results.

No.	Electrolyte Concentration *X*_1_	Current Density *X*_2_	Electrode Distance *X*_3_	RE(%) *Y*
1	0	1	−1	51.18
2	1	−1	0	57.43
3	1	0	−1	60.19
4	0	0	0	64.93
5	0	−1	1	52.93
6	−1	1	0	50.02
7	1	1	0	55.66
8	0	0	0	64.06
9	0	0	0	66.55
10	0	0	0	66.32
11	0	−1	−1	55.43
12	1	0	1	52.48
13	0	1	1	42.05
14	0	0	0	67.11
15	−1	0	1	53.66
16	−1	0	−1	59.85
17	−1	−1	0	60.04

**Table 3 polymers-16-00340-t003:** Result of ANOVA for the RE of MBC.

Source	Sum of Squares	DF	Mean Square	*F*-Value	*p*-Value	Remarks
Model	762.02	9	84.67	57.86	<0.0001	**
A	0.60	1	0.60	0.41	0.5425	
B	90.59	1	90.59	61.90	0.0001	**
C	81.47	1	81.47	55.67	0.0001	**
AB	17.02	1	17.02	11.63	0.0113	*
AC	0.58	1	0.58	0.39	0.5498	
BC	10.99	1	10.99	7.51	0.0289	*
A^2^	15.68	1	15.68	10.71	0.0136	*
B^2^	274.69	1	274.69	187.70	<0.0001	**
C^2^	225.58	1	225.58	154.14	<0.0001	**
Residual	10.24	7	1.46			
Lack of fit	3.91	3	1.30	0.82	0.5454	
Pure error	6.33	4	1.58			
Cor total	772.26	16				

Note: R^2^ = 0.9867, R^2^_Adj_ = 0.9697, R^2^_Pred_ = 0.9062; * Significant, *p* < 0.05, ** Highly significant, *p* < 0.01.

## Data Availability

Data are contained within the article.

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
