# Peer review of "Treatment of Monochlorobenzene from Polymers Process through Electrochemical Oxidation"

_polymers, 2024, doi:10.3390/polym16030340_

Round 1

Reviewer 1 Report

Comments and Suggestions for Authors

In this manuscript, the authors have presented the results of the optimization of the conditions for the electrochemical oxidation system deigned for the monochlorobenzene oxidation. The influence of experimental parameters, e.g., electrolyte content, electrode dimensions and distance between them, wee considered and modelled using a Design-Expert 10.0.1 software. The results were discussed in terms of the processes that influence target electrode reactions and efficiency of the oxidation.  The topic of investigations seems interesting and deserves publication though it is hardly related to the polymers. The results are presented in a clear and convincing way. Nevertheless, some more attention to the details of experiment are required. First and foremost, the choice of the variable parameters and of the range of their variation should be explained. The materials used for electrode manufacture should be characterized by conventional methods or referred to appropriate information provided by supplier. The reactions that  take part on he electrodes should be presented with some arguments like polarization curves etc. to prove the nature of the products used. Thus, the cathodic reaction of oxygen can result in formation of appropriate anion radical but not of hydrogen peroxide. Direct reduction of C-Cl bond is also probable. Then, the stability of the electrolysis including the assessment of the electrode lifetime and possible changes caused by the intermediates collected on the electrodes (passivation, chemisorption) should be considered and discussed.

technical notes:

Abstract: it is recommended to shorten first part confirming urgency of research made

Keywords: it is recommended to remove or modify the key words “polymer intermediate”

Introduction: general description of the polymers (first paragraph, pages 1 and 2) has a little relation to the topic of investigation and should be reduced

Introduction: the number of acronyms is exhausting and can be easily reduced if they are mentioned few times – please reconsider the necessity and remove some of them

Section 2.1 – the size of the electrodes and the distance between them should be mentioned here, not in the 2.2.1 section

The manufacture of the electrode materials or the protocol of its production should be added

Section 3.1 – it is not clear why only Ti/Ti4O7 material was characterized while other ones did not contain similar parameters.

Section 3.2 – please specify what electrode square is mentioned – geometric one or effective (real) surface square determined by electrochemistry for the  current densities present.

Section 3.3 lines 229-230 – leave the dimension of the concentration at the last value, do not repeated it with each concentration.

Section 3.4 line 251 – leave the dimension of the RE at the last value, do not repeated it with each percentage, the same in Page 10, line 346

Section 3.5 – please remove the distance from the section title.

Section 3.5 line 262 – leave the dimension of the distance at the last value, do not repeated it with each distance (should be: 3.0, 4.0, 5.0 and 6.0 cm). The same in line 276

Table 2 – Please remove the dimension of the variables form the column titles (dimensionless values are used)

Section 3.6.2. – please change the “interactive effect” by “interaction effect”

Comments on the Quality of English Language

Some technical problems with the verb tenses

Author Response

Thank you again for your affirmation to our work. We look forward to hearing from you.

Thank you and best regards.

Yours sincerely,

Reviewer 2 Report

Comments and Suggestions for Authors

In this study, the electrochemical oxidation system was established to treat- ment monochlorobenzene (MCB) waste gas. The effect of single factor such as anode materials, cathode materials, electrolyte concentration, current density and electrode distance on removal efficiency (RE) of MCB gas were the first study. I suggested a minor revision.

- In the abstract, please add the several sentences to describe significance of the Monochlorobenzene with Electrochemical Oxidation.

- The English language through article should be carefully polished.

- The introduction section was improved with novel references including  monochlorobenzene detection methods. 

The authors used Polymer Intermediate which is preferred due to its good electrical conductivity, chemical stability with high surface area. But, the importance of these novel Polymer Intermediates were not indicated in manuscript. Especially, the section of introduction is very important to indicate these properties.

- Error bars should be included in Figure 6.

- The validation conditions of  monochlorobenzene (MCB) waste gas were investigated in detail.

- The experimental section was explained in detail especially  monochlorobenzene (MCB) treatment. 

Comments on the Quality of English Language

The English language through article should be carefully polished.

Author Response

Thank you again for your affirmation to our work. We look forward to hearing from you.

Thank you and best regards.

Yours sincerely.

Round 2

Reviewer 1 Report

Comments and Suggestions for Authors

I am fully satisfied with the amendments and changes made in the manuscript in accordance with the reviewer's report. It can be accepted in present form